# DMSO induces major morphological and physiological alterations in zebrafish embryos

**Geyse Gomes, Allaina Christina de Sousa Andrade, Paloma de Carvalho Vieira, Murilo Nespolo Spineli[iD], Manoel Luis Costa, Claudia Mermelstein[iD]** *

Laboratório de Diferenciação Muscular, Instituto de Ciências Biomédicas, Universidade Federal do Rio de Janeiro, Rio de Janeiro, Brazil

* mermelstein@ufrj.br

## Abstract

Dimethyl sulfoxide (DMSO) is a polar aprotic solvent that dissolves both polar and nonpolar compounds and is miscible in a wide range of organic solvents as well as water. These characteristics render DMSO one of the most widely used solvents in pharmaceutical industry, as well in basic biomedical research, such as in developmental biology studies. The analysis of the effects of DMSO during the development of zebrafish larvae is particularly important since zebrafish is one of the most studied vertebrate models in developmental biology and in toxicology. It has been reported that DMSO at concentrations up to 1% are safe to be used in zebrafish embryo developmental toxicity assays, but fundamental questions such as whether and how different concentrations of DMSO affect the morphology and physiology of zebrafish have not been investigated. Therefore, our study focused on the analysis of DMSO treatment during zebrafish development using high resolution optical microscopy and real-time video-microscopy of the whole embryo, as well as focusing on some organs and tissues. Our results show that concentrations above 5% of DMSO are lethal to embryos, whereas concentrations between 1–4% induce different morphological and physiological alterations in embryos. The alterations observed include up-curved tail, heart beating frequency, heart edema, somite size, myofibril alignment, melanocyte size, notochord and swim bladder morphology. These results show for the first time a detailed analysis of the major effects of the widely used solvent DMSO during zebrafish development and call for attention in the use of DMSO in basic and applied research.

## Introduction

Zebrafish (*Danio rerio*) is a powerful and well-established vertebrate model for developmental biology studies. The advantages of this model include high fecundity (hundreds of embryos in a single clutch), external fertilization, rapid development, transparency of the developing embryo allowing live imaging at the cellular, tecidual

**Data availability statement:** All relevant data are within the manuscript and its Supporting Information files.

**Funding:** This work was supported by Conselho Nacional de Desenvolvimento Científico e Tecnológico (CNPq, funding number 302961/2021-6 to C.M., 308192/2021-4 to M.L.C.) and Fundação de Apoio à Pesquisa do Estado do Rio de Janeiro (FAPERJ, funding number E-26/203.930/2024 to C.M., E-26/204.077/2024 to M.L.C.). There was no additional external funding received for this study.

**Competing interests:** The authors have declared that no competing interests exist.

and organismal levels, easy genetic and experimental manipulation, fully sequenced genome, presence of most of the major organ systems of vertebrates and metabolic characteristics similar to humans [1,2]. These characteristics and advantages strongly increased the use of zebrafish not only in developmental biology studies but also in other fields, such as pharmacology and toxicology [3,4]. One problem that arose from this intense use of *D. rerio* was the concomitant increase in the use of dimethyl sulfoxide (DMSO), a commonly used solvent for dissolving compounds that are not miscible in water [5,6].

DMSO is widely used as a control or as part of the media component. Several studies report the effects of DMSO in zebrafish [7–10]. Christou *et al*. [7] showed that DMSO affects larval zebrafish behavior at a concentration of ≥0.55%, with additive and interaction effects when combined with two positive controls for larval locomotion, flutamide and perfluorooctanesulfonic acid (PFOS). Kyeongnam and Sung-Eun [10] reported that DMSO did not show acute toxicity on zebrafish embryos at the concentrations of 0.1%, whereas synergistic acute toxicity was found in the simultaneous treatment of DMSO and vanadium. The combined toxicity they observed delayed the development of zebrafish embryos and caused pericardial edema. Confirming these previous results, Hoyberghs and colleagues [9] reported that DMSO at concentrations up to 1% are safe to be used in zebrafish embryo developmental toxicity assays. Hedge *et al*. [8] reported that DMSO altered zebrafish locomotor activity and behavior following developmental exposure at concentrations as low as 0.5%. The collection of these results shows that DMSO can induce different levels of morphological and behavioral alterations in zebrafish larva depending on its concentration, on the duration of treatment, in its combination with other drugs, on the tissues/organs analyzed, and on the methods/techniques used to analyze its effects, among other parameters.

The molecular mechanism of action of DMSO has been dissected by Gurtovenko and Anwar [11], who showed its interaction with phospholipid membranes by three distinct modes of action, each over a different concentration range. At low concentrations, DMSO induces membrane thinning and increases fluidity of the membrane's hydrophobic core. At higher concentrations, DMSO induces transient water pores into the membrane. At still higher concentrations, individual lipid molecules are desorbed from the membrane followed by disintegration of the bilayer structure. Furthermore, Tunçer and colleagues [12] showed that low doses of DMSO (0.1–1.5%) can lead to the stabilization of Z-DNA (an alternate DNA form) which could be related to alterations in gene expression, differentiation, and epigenetic.

Accordingly, questions of whether, in which extent, and how different concentrations of DMSO affect the morphology and physiology of zebrafish have not been sufficiently investigated. Here we focused on the analysis of DMSO treatment during zebrafish development using high resolution optical microscopy and real-time video-microscopy of the whole embryo, as well as focusing on some organs and tissues. Our results show that DMSO induces major changes in zebrafish morphology, physiology and locomotion. We highlight the fact that care should be taken when using DMSO in zebrafish studies.

## Materials and methods

### Zebrafish maintenance and treatment

Adult wild-type *Danio rerio* fish were kept in system water at 28 ± 1°C on a 14:10 light/dark cycle in an animal facility at the Institute of Biomedical Sciences of the Federal University of Rio de Janeiro, Brazil according to standard procedures [13]. Animals were handled according to Institutional Animal Care and Use Committee protocols under the number 036/21. Embryos and larvae were collected, dechorionated and the treatment started at 24 post fertilization (hpf) with increasing concentrations of Dimethyl Sulfoxide (DMSO 1, 2, 3, 4 and 5%, v/v, from Sigma-Aldrich, code D8418, 99.9% purity) up to 72 hpf. DMSO was diluted in E3 solution (5 mM NaCl, 0.17 mM KCl, 0.33 mM $CaCl_2$, and 0.33 mM $MgSO_4$ in distilled water, pH 7.2, which was made in our lab) before each experiment. Six-well plates were used for the experiments. For experiments with 72 hpf, 10 embryos were distributed per well containing 2 mL of solution. Ten technical replicates (embryos with the same treatment within the same plate) and three biological replicates (embryos with the same treatment but performed at a different day and mating) were done (n = 30). Four biological replicates were used for experiments with 7 dpf. Zebrafish embryos and larvae kept in E3 solution without DMSO were considered negative control.

### Bright field microscopy, imaging and quantification

Live embryos were visualized in a Zeiss Axiovert 100 inverted microscope (Carl Zeiss, Germany). Images were acquired with an Olympus DP71 high-resolution camera (Olympus, Japan). Images were analyzed and processed with the ImageJ software, based on the public domain NIH Image program (developed at the U.S. National Institutes of Health and available on the Internet at http://rsb.info.nih.gov/nih-image/). Plates were prepared using Adobe Photoshop (Adobe Systems Incorporated, USA). The following morphological features of control and DMSO-treated embryos were quantified: embryo's caudal length, embryo's thickness, pericardium area, yolk area, yolk-caudal length ratio, curvature index (CI) and somite size (measured in the middle of somite). CI was calculated as the ratio between Feret diameter and tail length (a straight line has a CI of 1). All raw quantitative data for each measured parameter can be found in S1 Table.

### Immunofluorescence microscopy

Analysis of zebrafish embryos by fluorescence microscopy was performed according to previously described procedures [14]. Dechorionated zebrafish embryos (control and DMSO-treated) were fixed in 4% paraformaldehyde in PBS for 1 hour at room temperature. They were permeabilized with Triton X-100 0.5% in PBS (PBS-T) for 30 minutes and incubated for 1 hour at 37°C with primary monoclonal antibody anti-vinculin (code # V4505, Sigma-Aldrich, diluted 1:200 in PBS/T). After a 30-minute wash with PBS-T, embryos were stained with Alexa Fluor 488-anti-mouse secondary antibody (ThermoFisher, diluted 1:200). Embryos were washed for 30 minutes with PBS and incubated with Alexa Fluor 647-Phalloidin (diluted 1:200 in PBS) for 1 hour at 37°C. Then, embryos were washed for 30 min with PBS and incubated with 0.1 µg/mL of DAPI (Thermofisher, diluted 1:2000 in NaCl) for nuclei stain. Embryos were washed for 5 minutes with NaCl and mounted on 24×60-mm glass coverslips (with spacers) using Prolong Gold (Thermofisher). Embryos were observed in a DSU Spinning Disk Confocal microscope (Olympus, Japan). Control experiments with only secondary antibodies showed only a faint background staining (data not shown). Image processing (brightness, contrast adjustments and somite size quantification) was performed using Fiji software [15] and figure panels were mounted with Adobe Photoshop software (Adobe Systems Inc., USA).

### Functional analysis

Heartbeat of 72 hpf embryos were recorded in control and in DMSO-treated embryos, as previously described [16]. To analyze the heart beating for each experimental condition, zebrafish embryos were observed in a Zeiss Axiovert 100 inverted microscope (Carl Zeiss, Germany) and hearts were recorded with an Olympus DP71 high-resolution camera (Olympus, Japan). Videos were analyzed and processed with the ImageJ software.

Larvae with 7 dpf were placed in 35-mm culture dishes filled with 2 mL of E3 solution. Larvae movements (10 larvae per dish) were recorded with a cell phone (4k, 30fps, 3x optical zoom) for 10 minutes. Movies were analyzed by the public domain program ImageJ and the swimming area of larvae was quantified. We subtracted the background of each frame (background of all the movie images was generated by an average Z-projection) and we used a Z-projection of the minimal values to generate the final image with all the positions. The dark areas in the final projections represent all the positions the embryos occupied in the movie and they were quantified by thresholding with the same value for all conditions.

## Statistical analysis

Statistical analysis was carried out using GraphPad Prism software version 9. The results of at least three independent experiments were compared. Statistical analysis of data related to the quantification of cells was performed with One-way ANOVA followed by Tukey's multiple comparison test, $*p < 0.05$, $**p < 0.01$, $***p < 0.001$, $****p < 0.0001$.

## Results

DMSO is a universal solvent for compounds that are not miscible in water. An analysis of the number of papers published in the Europe PMC (https://europepmc.org/) database using the descriptor "DMSO" resulted in 484,992 articles published (data of search June 01, 2025) in a period that spanned the years 1963–2025. Beginning in 2007, there was an exponential increase over time in the number of publications with DMSO. Remarkably, in only one year (2022) more than 45,000 articles with the descriptor DMSO were published, confirming that DMSO has been widely used during the last 60 years. Analysis of the content of these articles showed that DMSO is used in basic biomedical research, cryopreservation, pharmaceutical industry, as well as in developmental biology studies. A search for papers in the Europe PMC using both "DMSO" and "developmental biology" resulted in 15,863 (data of search June 01, 2025) in a period that spanned the years 1971–2025.

The indiscriminate use of DMSO in developmental biology research led us to ask whether this solvent could have unwanted effects in zebrafish (*Danio rerio*) at the cellular, tecidual, and whole organism levels. We tested concentrations ranging from 1 to 5% of DMSO, besides the control without DMSO. First, we analyzed the survival rate of embryos after 24 and 48 h of treatment with 1–5% DMSO (Fig 1A-C). Survival rates were similar to control (untreated) embryos, except for 5% DMSO, in which no embryos survived after 48 h of treatment (Fig 1C). Analysis of the overall morphology of treated embryos under bright field optical microscopy showed that DMSO induced a dose-dependent up-curved phenotype in embryos treated with 3–5% DMSO for 24 hs (Fig 1A). Embryos treated with DMSO for 48 h showed an up-curved phenotype with 2–5% DMSO (Fig 1B). Importantly, all embryos died after 48 h of treatment with 5% DMSO, and 98% of these dead embryos were up-curved. These results show that the effects of DMSO on zebrafish embryos are concentration and time/duration dependent. We quantified the percentage of untreated and DMSO-treated embryos that have an up-curved phenotype and found that 6.6% of the embryos were curved in 1% DMSO, 16.6% in 2% DMSO, 36.6% in 3% DMSO, 66.6% in 4% DMSO, and 96.6% in 5% DMSO after 24 h of treatment, whereas 66.6% of the embryos were curved in 2% DMSO, 73.3% in 3% DMSO, 93.4% in 4% DMSO, and 96.6% in 5% DMSO after 48 h of treatment (Fig 1C).

Analysis of DMSO-treated embryos showed morphological alterations, other than up-curved tail, in several structures and organs, including the heart, somites, notochord and swim bladder. We then decided to further investigate and quantify these morphological alterations. All embryos treated with 5% DMSO die after 48 hs (Fig 1C). Embryo's length decreased after treatment with 2–4% of DMSO, whereas embryo's thickness decreased after treatment with 4% DMSO (Fig 2A-B). Analysis of the heart of zebrafish embryos showed a decrease in the pericardial area after 3% and 4% DMSO treatment (Fig 2C), and cardiac edema in 7% of embryos treated with 2% DMSO, 17% with 3% DMSO, and 19% with 4% DMSO after 48 h of treatment. Not only was the heart morphology altered but the heart beating rate decreased in embryos treated with 3 and 4% DMSO (Fig 3A-F, S1 File). Yolk area and yolk/caudal length ratio increased after 3 and 4% DMSO

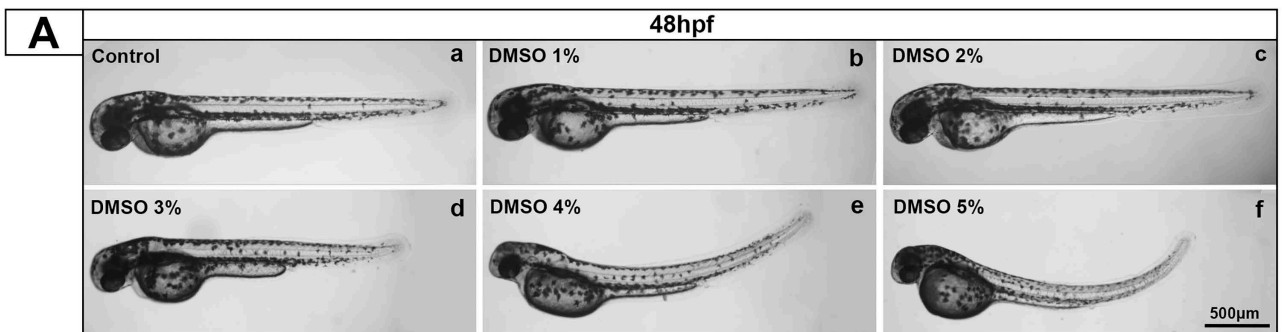

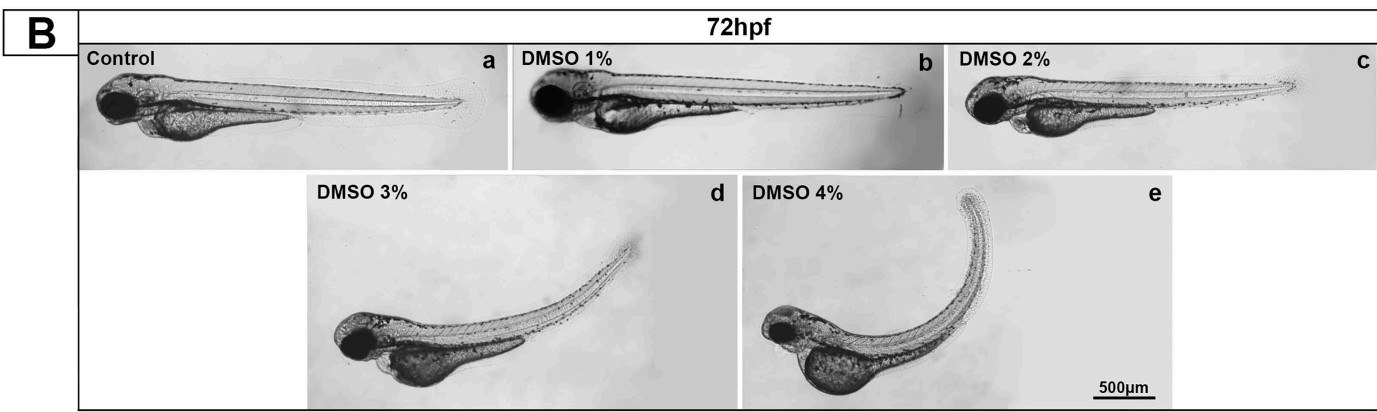

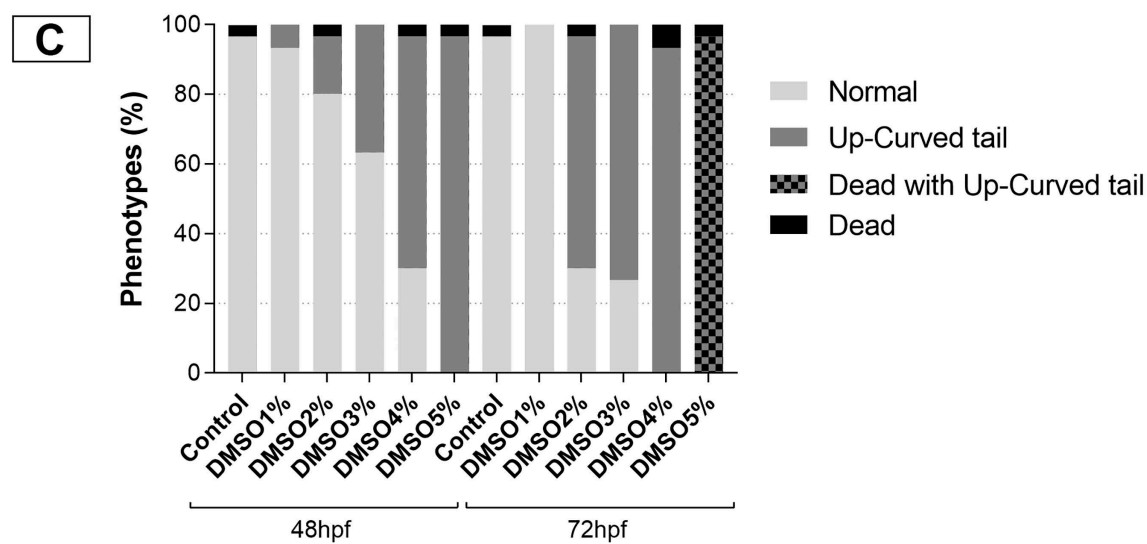

**Fig 1. Survival rate of zebrafish embryos after DMSO treatment.** Zebrafish embryos with 48 hs were treated with DMSO with concentrations ranging from 1 to 5% and survival rate was quantified after 24 hs and 48 hs of treatment. Note that nearly all embryos die after 48 hs of treatment with 5% of DMSO. Scale bars = 500 μm. Number of embryos analyzed per experimental group = 30.

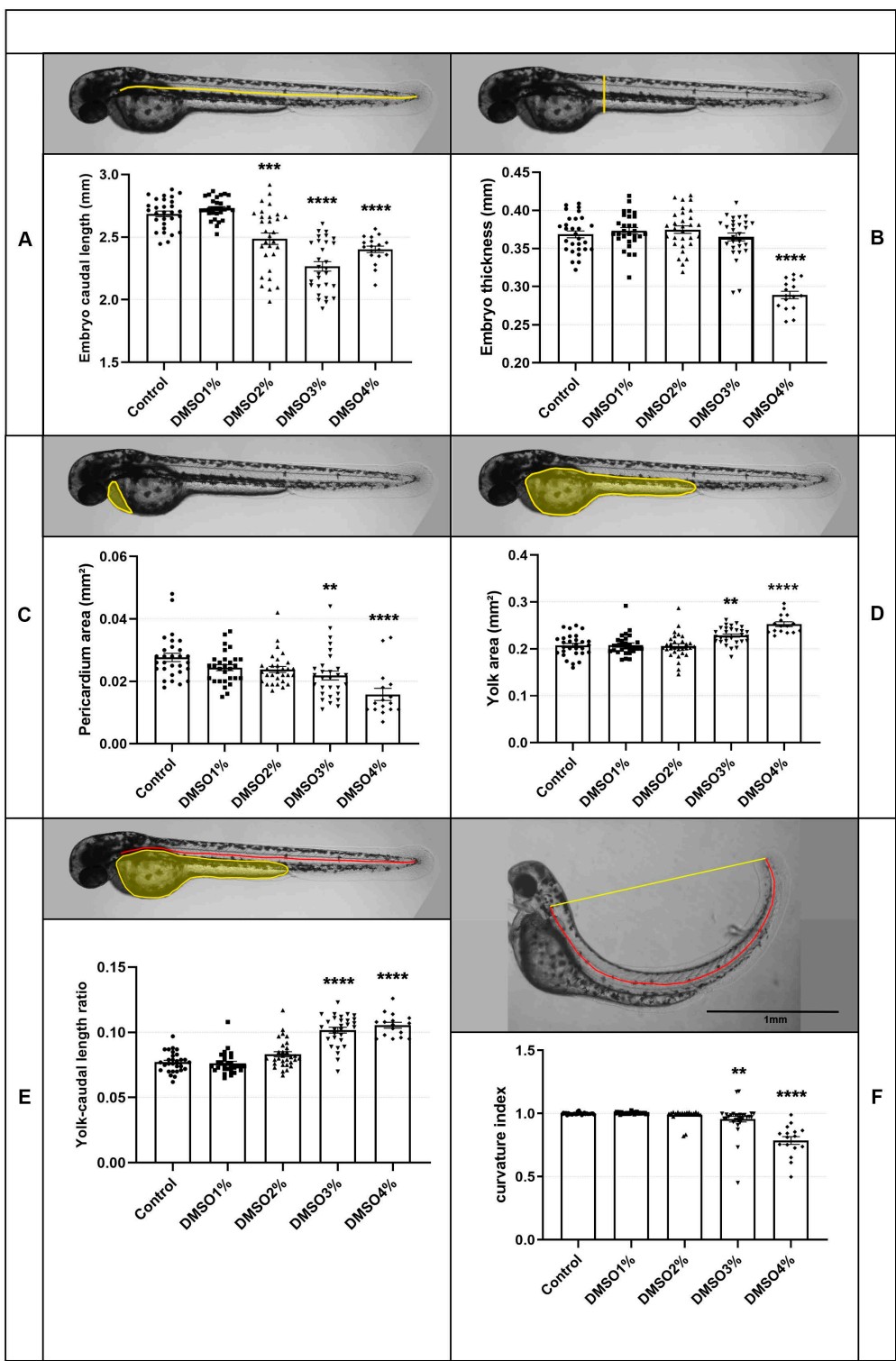

**Fig 2. DMSO alters the morphology of zebrafish embryos.** Zebrafish embryos with 48 or 72 hpf were treated with DMSO with concentrations ranging from 1 to 5%. Bright field images were acquired after 24 hs of treatment and quantification of several morphological parameters was performed. Scale bar = 1 mm. Number of embryos analyzed per experimental group = 30.

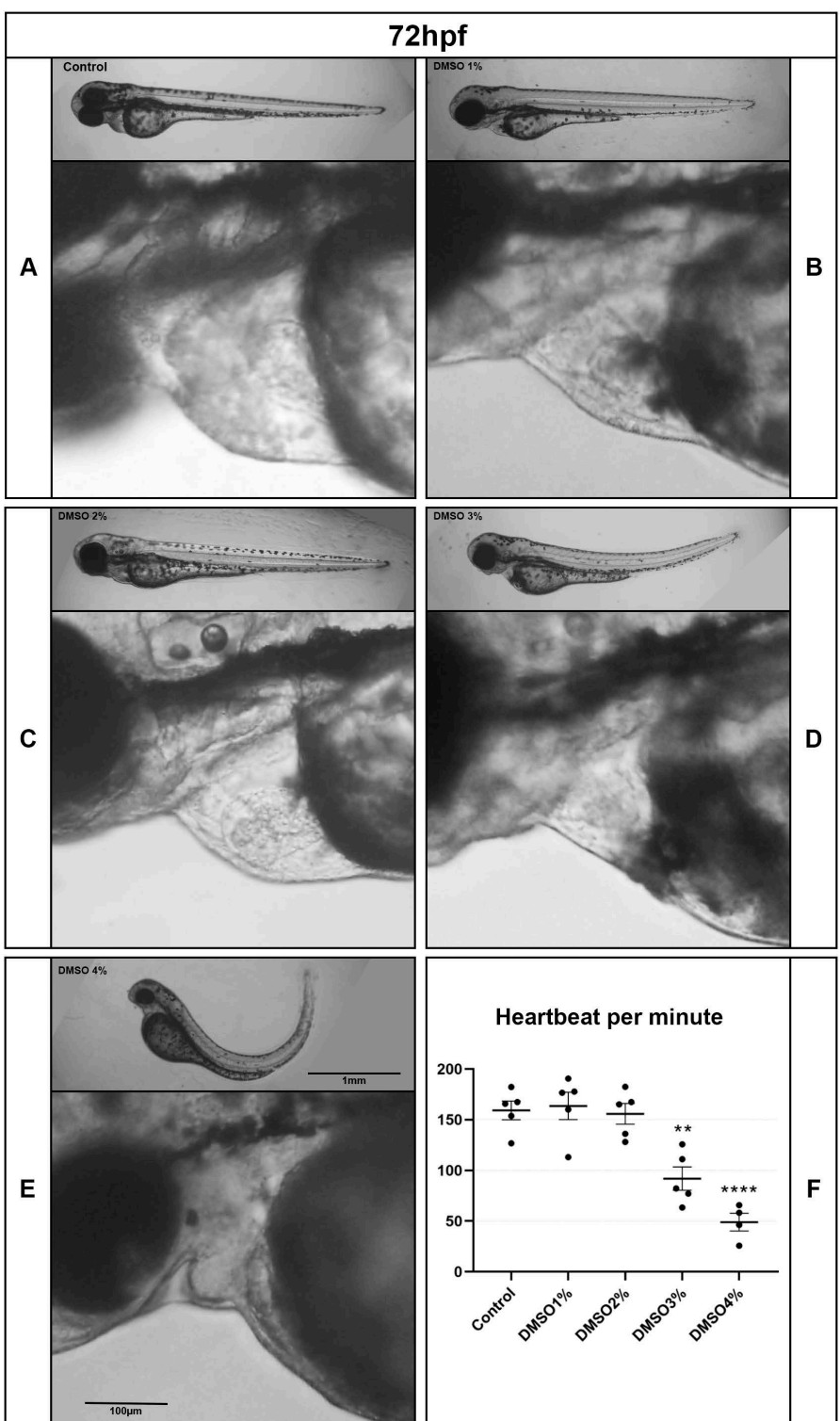

**Fig 3. DMSO induces cardiac edema in zebrafish.** Zebrafish embryos at 48 hpf were treated with DMSO with concentrations ranging from 1 to 5% and cardiac edema and heart beating were quantified. Scale bars = 1 mm and 100 μm. Number of embryos analyzed per experimental group = 30.

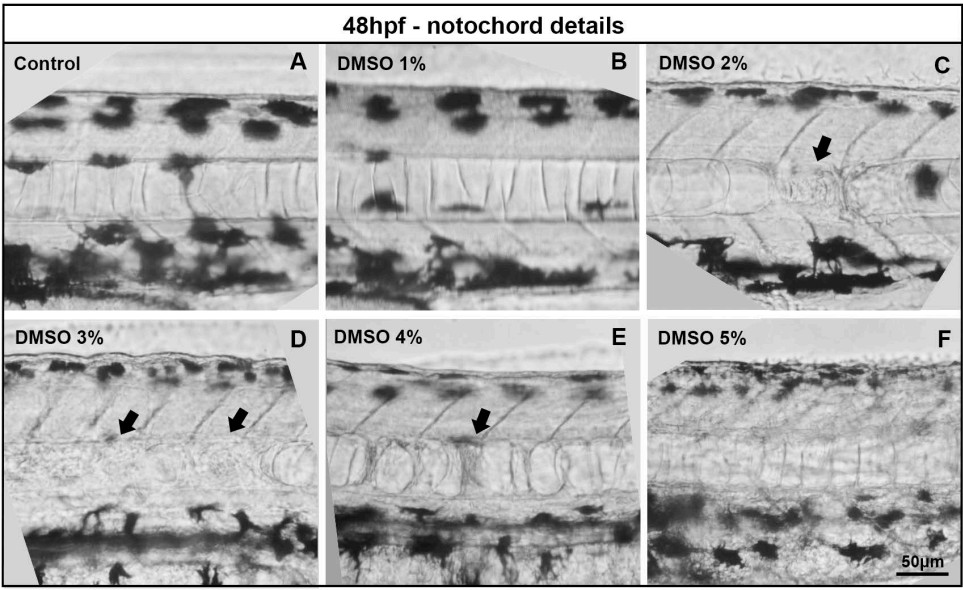

**Fig 4. DMSO induced alterations in zebrafish notochord.** Zebrafish embryos with 48 hs were treated with DMSO with concentrations ranging from 1 to 5% and bright field images were acquired after 24 hs of treatment. Black arrows point to alterations in the notochord morphology of zebrafish treated with DMSO. Scale bar = 50 μm. Number of embryos analyzed per experimental group = 30.

(Fig 2D-E). Measurements of the curvature index of embryos showed a decrease after treatment with 3 and 4% of DMSO (Fig 2F).

Notochord morphology was also changed in embryos treated with 2–5% of DMSO (Fig 4). In normal zebrafish embryos, the notochord has a columnar arrangement of cells within the notochord sheath, which is an outer epithelial-like layer. DMSO treatment altered the notochord's columnar structure in a dose-dependent manner (Fig 4). Curiously, in the images of Fig 4 it is possible to observe that DMSO induces a dose-dependent decrease in melanocytes. Embryos treated with 5% DMSO showed smaller melanocytes, compared to untreated embryos (Fig 4).

Interestingly, analysis of zebrafish embryos under polarized light microscopy showed that somite's size and shape were also altered after DMSO treatment, which can be observed by birefringent images of embryo's skeletal muscles (top images in the middle column Fig 5). To further explore the effects of DMSO on muscle somites, we labeled zebrafish embryos with the F-actin specific probe Phalloidin (to stain striated myofibrils) and DAPI (to stain the nuclei). A more detailed observation showed that striated myofibrils were misaligned after DMSO exposure and actin was abnormally concentrated near the septa (Fig 6). Embryos showed a significant dose-dependent decrease in the size of somites with DMSO treatment (Fig 7). Somites in control untreated embryos measured 81.1 μm (± 1.8 SEM) and in DMSO-treated embryos somites measured between 60.5–78.9 μm in thickness (Fig 7D).

We also analyzed the effects of DMSO in the movement of zebrafish larvae. Control and DMSO-treated larvae with 7 dpf were placed at the center of a 35-mm culture dish and their movements were recorded. Embryos treated with 1% and 2% DMSO showed a significant dose-dependent decrease in motility (S2 File). Embryos did not survive after 7 days in doses higher than 2% DMSO. Next, we tested whether concentrations below 1% DMSO could lead to alterations in the movement of zebrafish larvae. Larvae were treated with 0.1%, 0.3%, 0.5%, 1% and 2% DMSO and their movements were recorded for 10 minutes. The total swimming area of larvae was quantified, and no significant alterations were found (S3 File), although a dose-dependent tendency of decrease in locomotion was observed. We cannot discard the possibility that DMSO in concentrations below 1% could alter zebrafish larvae morphology. Further studies are necessary to unravel this possibility.

 

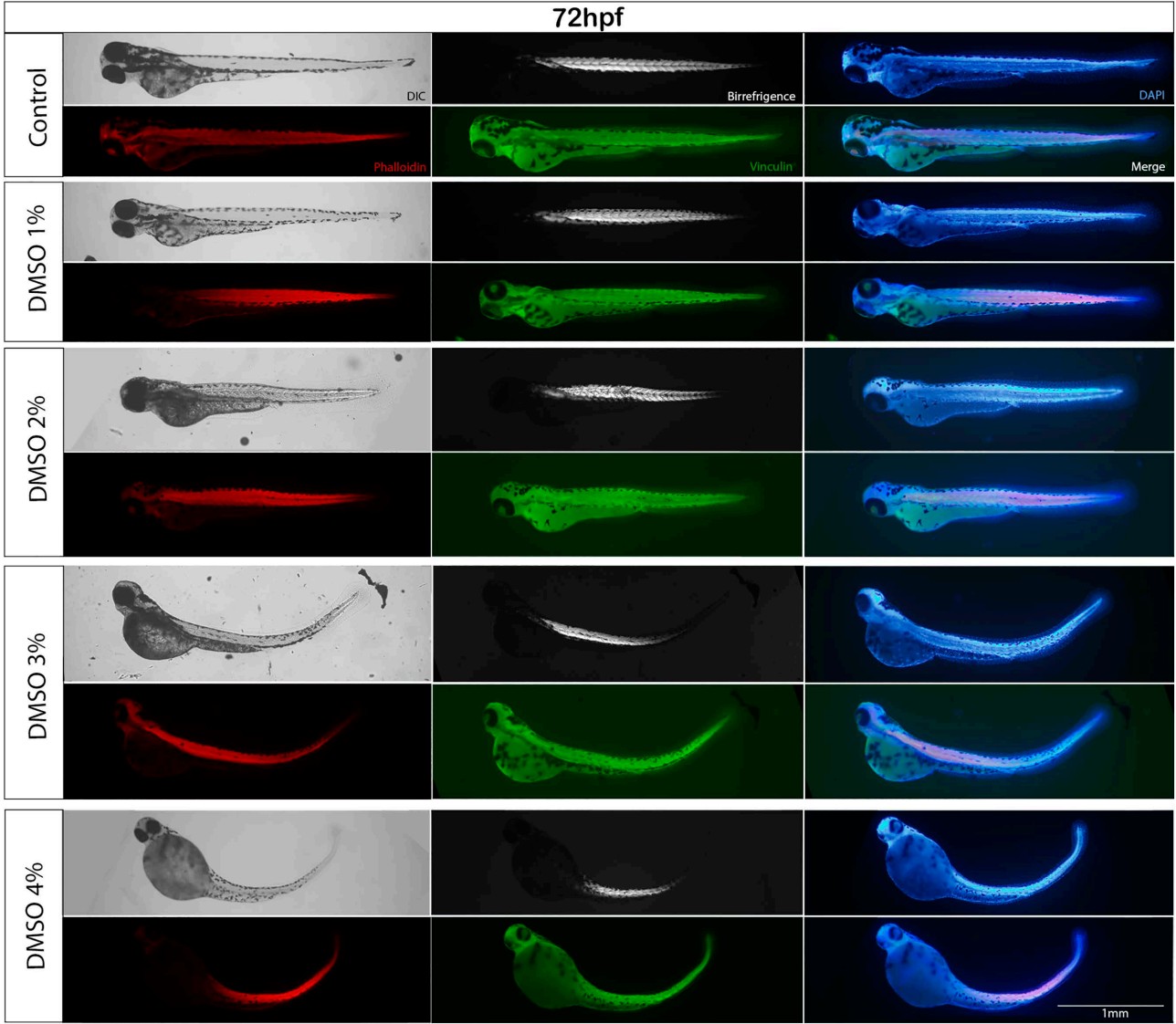

**Fig 5. DMSO alters muscle somites.** Zebrafish embryoswere observed under polarized and fluorescent microscopy. Embryos were labelled for F-actin with Phalloidin (red), for vinculin (green) and with DAPI (blue). Somites size and shape were altered after DMSO treatment, which can be observed by the birefringent images of embryo's skeletal muscles (top middle images in the middle column). Scale bar = 1 mm. Number of embryos analyzed per experimental group = 30.

Next, we decided to compare the effects of short term DMSO treatment (24 and 48 hs) versus long term DMSO treatment (6 days) of zebrafish embryos. Remarkably, all embryos died after 6 days of treatment with higher doses of DMSO (more than 3% DMSO). No up-curved phenotype was observed in embryos treated with lower doses (less than 2%) of DMSO for 6 days (Fig 8), but these low DMSO concentrations induced significant dose-dependent changes in embryos, such as an increase in body thickness and decrease in swim bladder size (Fig 9). No apparent changes were observed in somite morphology after long term DMSO treatment (Fig 10).

The collection of the above-described results shows that DMSO can significantly, in a dose-dependent manner, alter the morphology and physiology of zebrafish embryos, including changes in embryo's size and curvature, heart beating

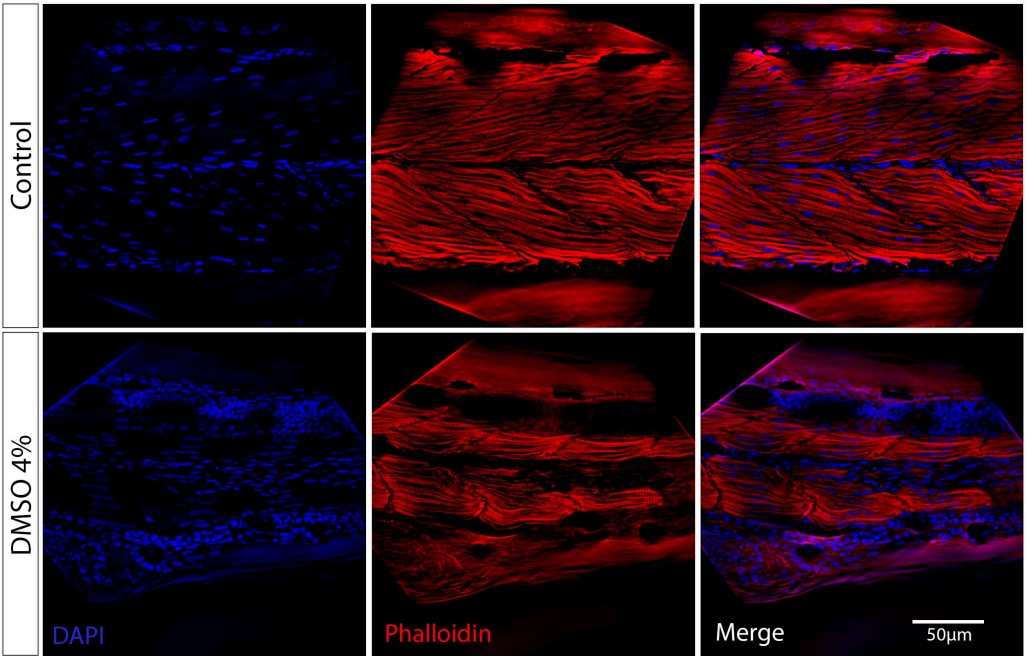

**Fig 6. DMSO treatment interferes with myofibril alignment.** Confocal analysis of zebrafish larvae at 48 hpf stained with the F-actin specific probe Phalloidin (red) and DAPI (blue). In control embryos, aligned striated myofibrils can be observed, whereas misaligned myofibrils are seen in 4% DMSO treated embryos. Scale bar = 50 μm. Number of embryos analyzed per experimental group = 30.

frequency, somite size and myofibril alignment, melanocyte size, and the morphology of heart, notochord and swim bladder (Table 1).

## Discussion

Here we analyzed the effects of DMSO during zebrafish development. Our results show that concentrations above 5% of DMSO are lethal to embryos, whereas lower concentrations (1–4% DMSO) induce dose-dependent morphological and physiological alterations in embryos. The alterations observed in live 24-hpf embryos treated with 1–4% DMSO for 48 hs were up-curved tail (in 3–4% DMSO), embryo size (in 2–4% DMSO), heart beating frequency (in 3–4% DMSO), pericardium area (in 3–4% DMSO), somite size (in 2–4% DMSO), yolk area (in 3–4% DMSO), smaller melanocytes (in 2–4% DMSO) and notochord morphology (in 2–4% DMSO). These alterations highlight major concerns for using DMSO as a solvent for *in vivo* studies. The relevance of our study is that zebrafish is one of the most studied vertebrate models in developmental biology and in toxicology and several of these studies use DMSO to dissolve polar and nonpolar compounds.

Remarkably, our results show that 96% of zebrafish embryos were up-curved after treatment with 4% DMSO for 48 h and 98% with 5% DMSO for 24 and 48 h. The upward tail curvature (up-curved, curly up or U-shaped) phenotype is well-characterized for PDK2/ALK5 zebrafish mutants [17]. Mitochondrial dysfunction can alter ALK5 signaling pathway [18]. Interestingly, several studies have shown that DMSO affects mitochondrial physiology in different cell types [19–22]. We can hypothesize that DMSO interferes with mitochondrial physiology in zebrafish embryos causing altered ALK5 signaling, which could lead to an up-curved phenotype with further effects depending on the cell type/tissue/organ. Why are mitochondria affected by DMSO? Gurtovenko and Anwar [11] suggested that the interaction of DMSO with membranes can induce membrane thinning, increase in membrane fluidity, and induce transient water pores into the membrane.

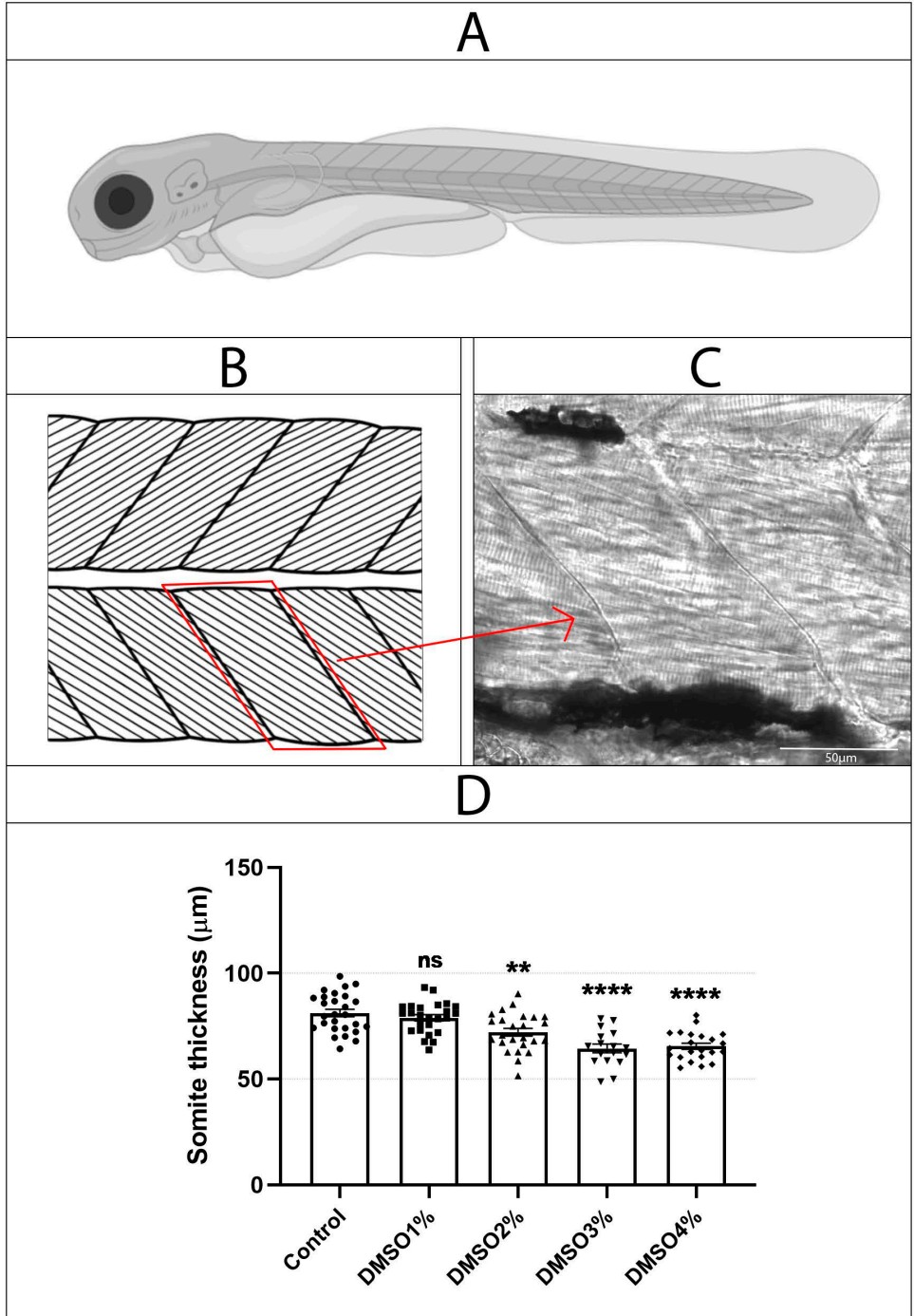

**Fig 7. DMSO induces a reduction in somite's size.** Zebrafish embryos (A) were analyzed under Differential Interference Contrast microscopy (DIC) in a Disk Confocal microscope and somites (B) were measured in their middle line (C). Somites' thickness significantly decreases after treatment with 2%, 3% and 4% DMSO (D). ns = non-significant. Number of embryos analyzed per experimental group = 30.

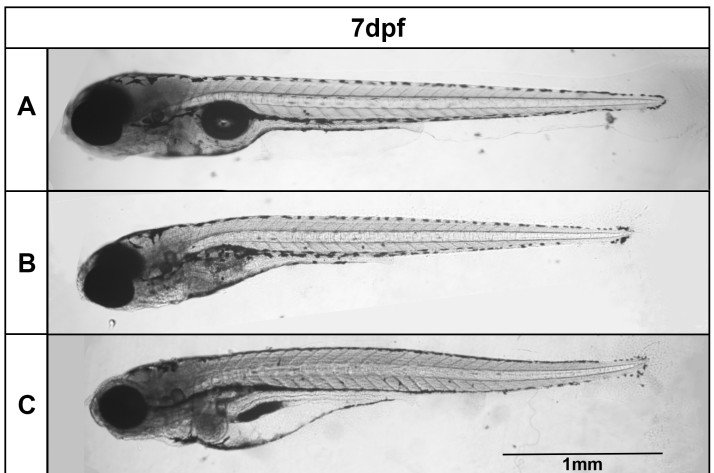

**Fig 8. Long term DMSO treatment does not induce an up-curved phenotype.** Zebrafish embryos were treated with DMSO for 6 days and no changes in body curvature were observed under bright field microscopy. Scale bar = 1 mm. Number of embryos analyzed per experimental group = 40.

These changes in the plasma membrane allow DMSO to enter the cell and once inside it can induce water pore formation in mitochondria, leading to altered mitochondrial physiology (Fig 11). This hypothesis could be tested with a detailed analysis of possible alterations in mitochondrial function and morphology after DMSO treatment. Distinct fluorescent probes are available to track oxygen consumption and changes in mitochondrial membrane potential, providing information about mitochondrial respiration and membrane permeability. Furthermore, DMSO effects on zebrafish embryos could also be investigated at the molecular and biochemical level through the analysis of the expression of genes and proteins involved in mitochondrial physiology and ALK5 signaling pathway, as well as markers of oxidative stress or apoptosis.

One of the most striking effects of DMSO in zebrafish embryos was alterations in muscle. Somite's size was reduced, and myofibril alignment was altered with DMSO treatment. These results could have an impact in the interpretation of the effects of specific DMSO-dissolved drugs used in several previous zebrafish muscle studies, particularly, in studies where no untreated control condition (without DMSO) is used in comparison with DMSO treatment. We can hypothesize that DMSO affects muscle by its interaction with phospholipid membranes of muscle cells. Alterations in the membranes of muscle cells (membrane thinning and increase in membrane fluidity) could have an impact in the intercellular communication between muscle cells and between muscle cells and the extracellular matrix present in the septa. Furthermore, the presence of actin aggregates (stained with Phalloidin) in somites near the septa of DMSO-treated zebrafish embryos strongly suggests that muscle fibers are strongly affected by DMSO. It has been shown that the presence of cytoplasmic actin aggregates in zebrafish somites correlates with reduced skeletal muscle function [23] (Sztal *et al*., 2015).

The only morphological parameter that increased with DMSO treatment was the area of the yolk. We can hypothesize that DMSO treated embryos were so deeply affected that they could not consume the yolk, as occurs during normal zebrafish development. Another possibility is that DMSO itself induces the increase in size of the yolk by an exacerbated accumulation of substances. These two hypotheses need to be tested in future experiments.

Remarkably, treatment with 1% and 2% DMSO for 6 days induced significant changes in embryos, such as an increase in body thickness and a decrease in swim bladder size. These results potentially contrast with the study of Hoyberghs and colleagues [9], who reported that DMSO at concentrations up to 1% are safe to be used in zebrafish embryo developmental assays. These differences could be explained by the developmental time when embryos were observed after treatment. We analyzed embryos after 6 days of 1% DMSO exposure and observed them after 120 hpf.

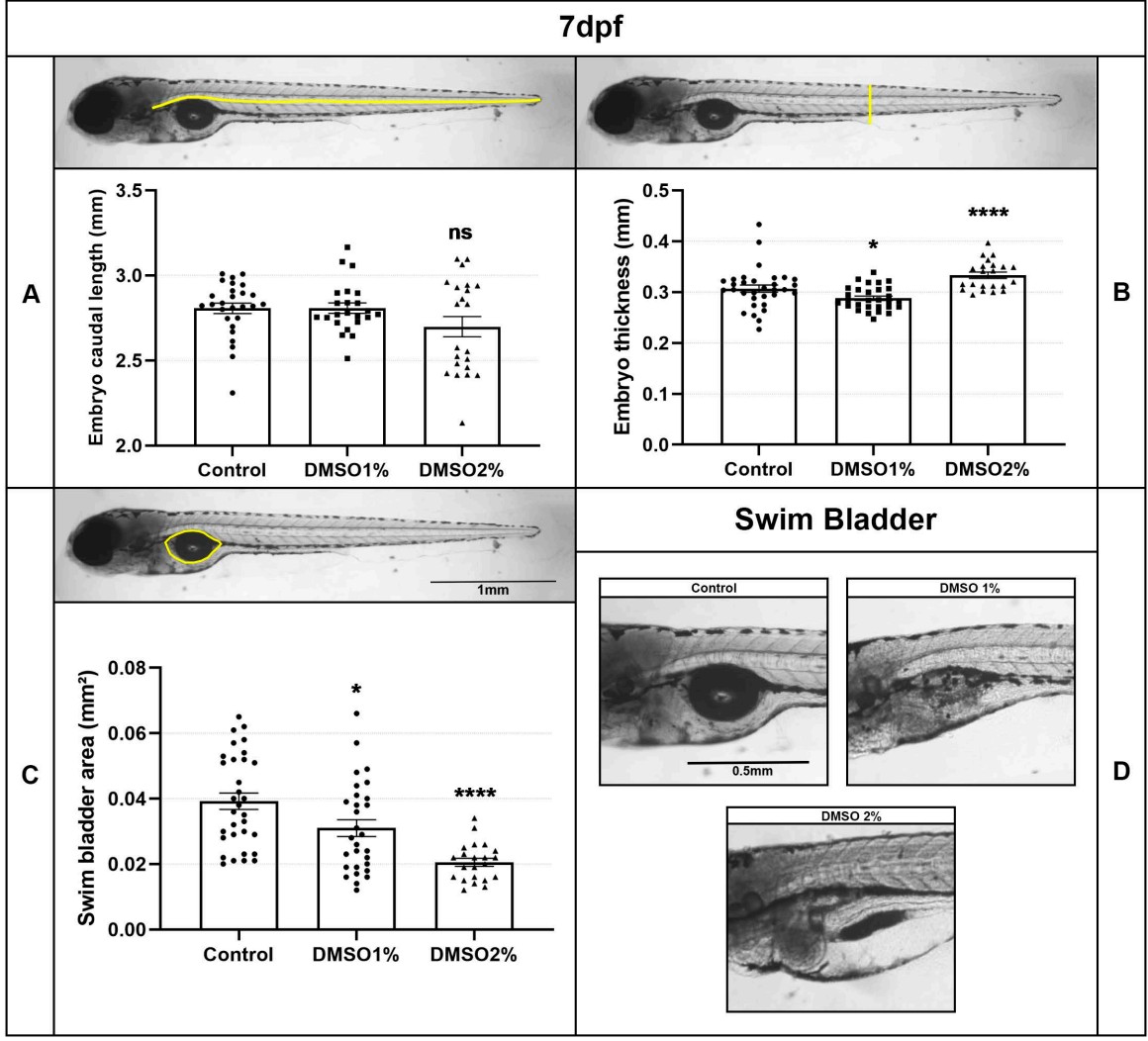

**Fig 9. Long term DMSO treatment induces changes in fish swim bladder.** Zebrafish embryos were treated with DMSO for 6 days and embryos caudal length, body thickness and swim bladder size were measured (A-C). Embryos' thickness increased while swim bladder size decreased after long term treatment. Images in D show the alterations induced by long term DMSO treatment in the swim bladder. Scale bars = 1 mm in C and 0.5 mm in D. Number of embryos analyzed per experimental group = 40.

Short term treatment with DMSO (24 and 48 hs) induced major changes in somite's size and shape, but unexpectedly somite morphology was unchanged by prolonged DMSO exposure (6 days). We can hypothesize that muscle cells, among other cells in zebrafish embryos, acquire resistance to DMSO over time. ATP-binding cassette (ABC) transporters, a ubiquitous family of integral membrane proteins, play a significant role in cellular resistance to drugs [24]. ABC transporters use ATP to translocate substrates across the plasma membrane removing them from the cell. It is possible that alterations in mitochondria by long term treatment with DMSO could alter ATP production and modulate muscle cell resistance to DMSO. Supporting this hypothesis, it has been shown that mitochondrial ATP fuels ABC transporter-mediated drug efflux in cellular chemoresistance [25].

Finally, the question of why DMSO is still widely used as part of the media if several studies have raised concerns about its use is related to its unique chemical properties. DMSO is a polar aprotic solvent that dissolves both polar and

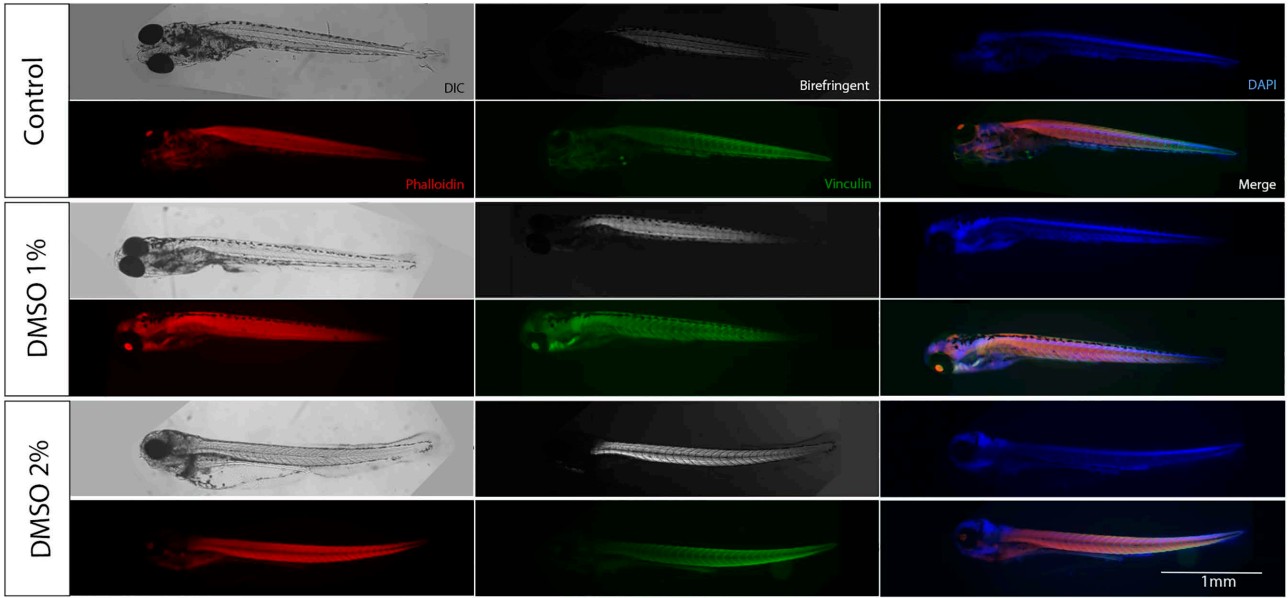

**Fig 10. Long term DMSO treatment does not alter zebrafish somites.** Zebrafish embryos were treated with DMSO for 6 days and observed under polarized and fluorescent microscopy. Embryos were labelled for F-actin with Phalloidin (red), for vinculin (green) and with DAPI (blue). Somites' size and shape were not altered after DMSO treatment, which can be observed by the birefringent images of embryo's skeletal muscles. Scale bar = 1 mm. Number of embryos analyzed per experimental group = 40.

**Table 1. Effects of DMSO on the morphology and physiology of zebrafish embryos.** DMSO can significantly, in a dose-dependent manner, alter the morphology and physiology of zebrafish embryos, including changes in embryo's size and curvature, heart beating frequency, pericardium area, somite size and myofibril alignment, melanocyte size, and morphology of notochord and swim bladder. (*) only the swim bladder area was quantified in 7 dpf zebrafish larvae treated with 2% DMSO, while all other measurements were quantified in 72 hpf zebrafish embryos treated with 4% DMSO. CI = curvature index.

| DMSO-treated zebrafish embryos phenotype | % of changes (control vs. 4% DMSO 72 hpf embryos) | values (mean ± SEM) |
|---|---|---|
| body linearity (CI) | 22% decrease | 0.99 ± 0.001 – normal<br>0.78 ± 0.030 – up-curved tai l(p < 0.0001) |
| caudal length (mm) | 0.7% decrease | 2.69 ± 0.023 – control<br>2.40 ± 0.025–4% DMSO (p < 0.0001) |
| body thickness (mm) | 22% decrease | 0.37 ± 0.004 – control<br>0.28 ± 0.005–4% DMSO (p < 0.0001) |
| yolk-caudal length ratio | 37% increase | 0.08 ± 0.001 control<br>0.11 ± 0.002–4% DMSO (p < 0.0001) |
| pericardium area (mm²) | 43% decrease | 0.03 ± 0.001 – control<br>0.02 ± 0.002 - 4% DMSO (p < 0.0001) |
| heartbeat (per min) | 70% decrease | 159.20 ± 9.282 control<br>48.98 ± 8.737 − 4% DMSO (p < 0.0001) |
| notochord cells | swollen | qualitative (not measured) |
| melanocytes | smaller | qualitative (not measured) |
| muscle myofibrils | misaligned | qualitative (not measured) |
| somite size (µm) | 19% decrease | 81.15 ± 1.761 – control<br>65.54 ± 1.437–4% DMSO (p < 0.0001) |
| swim bladder area (mm²) | 48% decrease (*) | 0.04 ± 0.002 – control<br>0.02 ± 0.001–2% DMSO (*) (p < 0.0001) |

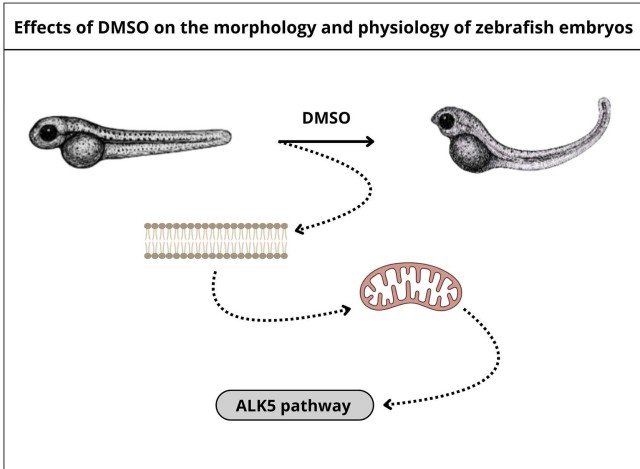

**Fig 11. Schematic representation of DMSO-induced changes in zebrafish embryos.** Our results show that 96% of zebrafish embryos were up-curved after treatment with 4% DMSO for 48 h and 98% with 5% DMSO for 24 and 48 h. The upward tail curvature phenotype is well-characterized for PDK2/ALK5 zebrafish mutants [17]. Mitochondrial dysfunction can alter ALK5 signaling pathway [18]. Our hypothesis is that DMSO alters plasma membrane permeability in ZF embryos cells, allowing DMSO to enter the cells and once inside them can induce water pore formation in mitochondria, leading to altered mitochondrial physiology and altered ALK5 signaling.

nonpolar compounds and is miscible in a wide range of organic solvents as well as water. Efforts should be made to find new solvents with lower toxicity for developmental biology studies. We suggest that zebrafish studies using DMSO as a solvent make efforts to use the lower concentration of DMSO that solubilizes the drug of interest and include an untreated control experimental condition where no DMSO is present. The comparison between DMSO-treated and untreated control is essential for evaluating the extent of DMSO toxicity in any *in vivo* experimental design.

Importantly, in specific scenarios the concentrations of 1–5% DMSO could still be used experimentally in zebrafish embryos despite the described known risks. For example, our data shows that heart beating frequency and pericardium area of 24-hpf zebrafish embryos were affected after 48 hs in the presence of 3–4% DMSO, but 2% DMSO could be used in heart-related studies even though these 3–4% DMSO could affect other organs or systems.

It is surprising that the overall effects of a drug used in such a high number of publications (475,368 articles from 1963–2025) had not been studied in detail before. The deleterious effects of DMSO (2–5%) that we described here are disturbing and require attention from the scientific community to further explore the alterations in specific cells, tissues and organs from not only zebrafish embryos, but other developmental biology model animals.

The novelty of our study resides in a comprehensive and detailed phenotypic characterization of the effects of DMSO across multiple zebrafish biological systems, including the heart, notochord, musculature, pigmentation, and motility. Our study describes systematic phenotypic mapping of the morphological and physiological effects of different concentrations of DMSO in zebrafish embryos and could be expanded to test other solvents that have similar properties to DMSO. The DMSO-associated phenotypes described here could be used as positive controls or reference models for developmental toxicity studies using zebrafish.

## Supporting information

**S1 Table. Raw quantitative data for each measured parameter of DMSO-treated zebrafish embryos.** All raw quantitative data related to the quantification of zebrafish embryos treated with DMSO or untreated are included in this table. (XLSX)

**S1 File. Supplementary Material 1–DMSO alters the heartbeat of zebrafish embryos.** Heartbeats of 48 hpf embryos were recorded in control and in DMSO-treated embryos. Alterations in the heartbeat of embryos can be observed after DMSO treatment. Number of embryos analyzed per experimental group = 30.
(AVI)

**S2 File. Supplementary Fig 2 and Supporting Information 3–DMSO decreases zebrafish motility at concentrations of 1% and 2%.** Larvae with 7 dpf were placed in 35-mm culture dishes filled with 2 mL of E3 solution. Movement of control and DMSO-treated larvae movements were recorded with a cell phone for 10 minutes. Three independent experiments were performed (N1, N2, N3) with control, 2% DMSO and 3% DMSO. The decrease in embryos motility was observed after DMSO treatment. Number of embryos analyzed per experimental group = 30.
(MOV)

**S3 File. Supplementary Material 4–DMSO does not alter zebrafish motility at concentrations below 1%.** Larvae with 7 dpf were placed in 35-mm culture dishes filled with 2 mL of E3 solution. Movement of control and DMSO-treated embryos were recorded with a cell phone for 10 minutes. Eight independent experiments were performed with control, 0.1% DMSO, 0.3% DMSO, 0.5% DMSO, 1% DMSO and 2% DMSO. No significant alterations in embryos motility were observed, although a dose-dependent tendency of decrease in locomotion was observed. Number of embryos analyzed per experimental group = 80.
(TIF)

## Author contributions

**Conceptualization:** Geyse Gomes, Allaina Christina de Sousa Andrade, Manoel Luis Costa, Claudia Mermelstein.

**Formal analysis:** Geyse Gomes, Allaina Christina de Sousa Andrade, Paloma de Carvalho Vieira, Murilo Nespolo Spineli.

**Funding acquisition:** Manoel Luis Costa, Claudia Mermelstein.

**Investigation:** Geyse Gomes, Allaina Christina de Sousa Andrade, Paloma de Carvalho Vieira, Murilo Nespolo Spineli.

**Methodology:** Geyse Gomes, Allaina Christina de Sousa Andrade, Manoel Luis Costa.

**Project administration:** Claudia Mermelstein.

**Supervision:** Geyse Gomes, Manoel Luis Costa, Claudia Mermelstein.

**Writing – original draft:** Claudia Mermelstein.

**Writing – review & editing:** Geyse Gomes, Allaina Christina de Sousa Andrade, Paloma de Carvalho Vieira, Murilo Nespolo Spineli, Manoel Luis Costa.

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
