## [Decision Letter · Decision Letter 0]

27 Jun 2025

PONE-D-25-27082DMSO induces major morphological and physiological alterations in zebrafish embryosPLOS ONE

Dear Dr. Mermelstein,

Thank you for submitting your manuscript to PLOS ONE. After careful consideration, we feel that it has merit but does not fully meet PLOS ONE’s publication criteria as it currently stands. Therefore, we invite you to submit a revised version of the manuscript that addresses the points raised during the review process.

We look forward to receiving your revised manuscript.

Kind regards,

Hai O. Xu

Academic Editor

PLOS ONE

 [This work was supported by Conselho Nacional de Desenvolvimento Científico e Tecnológico (CNPq, funding number 302961/2021-6 to C.M., 308192/2021-4 to M.L.C.) and Fundação de Apoio à Pesquisa do Estado do Rio de Janeiro (FAPERJ, funding number E-26/203.930/2024 to C.M., E-26/204.077/2024 to M.L.C.).].

Additional Editor Comments (if provided):

Reviewers' comments:

Reviewer's Responses to Questions

**Comments to the Author**

1. Is the manuscript technically sound, and do the data support the conclusions?

Reviewer #1: Yes

Reviewer #2: Yes

Reviewer #3: Yes

Reviewer #4: Yes

2. Has the statistical analysis been performed appropriately and rigorously? 

Reviewer #1: Yes

Reviewer #2: Yes

Reviewer #3: Yes

Reviewer #4: Yes

3. Have the authors made all data underlying the findings in their manuscript fully available?

Reviewer #1: Yes

Reviewer #2: Yes

Reviewer #3: Yes

Reviewer #4: Yes

4. Is the manuscript presented in an intelligible fashion and written in standard English?

Reviewer #1: Yes

Reviewer #2: Yes

Reviewer #3: Yes

Reviewer #4: Yes

5. Review Comments to the Author

Reviewer #1: Abstract & introduction: well written & adequate with logical flow.

Methodology:

- Well written, robust methodology, experimental setup is rigorous.

- The details of DMSO used in this study is not provided (i.e., source, grade, quality etc.) despite being the referenced chemical. The preparation of DMSO, how its quality is maintained throughout the course of the experiments are also not mentioned.

Results:

- Well written, explanations are thorough. Good quality images used.

- Figures and statistical analyses are comprehensive and effectively illustrate the dose dependent effect of DMSO

- Table 1 is qualitative. Could be further strengtened using quantified values if available (eg percentage of changes etc with concomittant p values)

Discussion:

- Well-written, interesting hypothesis on mitochondrial involvement, ALK5 signalling and membrane fluidity. Could be further strengtened by a schematic diagram sumarizing the hypothesized mechanism.

- Suggest to add future direction on how to validate these hypotheses.

- Suggest to comment a bit more on the long term effect of DMSO by adding details why somite morphology was unchanged by prolonged DMSO exposure (despite initial effects).

Reviewer #2: Revisión:

The paper entitles “DMSO induces major morphological and physiological alterations in zebrafish embryos” is unique research where the authors analyze the DMSO influence over the development of zebrafish larvae. Overall is an excellent article that show the importance of be aware of all the components that a media possess. I have some concenrs that I point out as follows:

- …”optical clarity of the developing embryo…”, the author could use the “transparency” to illustrate what they are trying to say.

- Eliminate the word “All” in the line…”All these characteristics…”.

- The second paragraph, which the authors present the DMSO problematic is kind weird. “…The most important characteristics of a solvent to be used as control, besides the ability to dissolve a given compound, is the lack of effects in the model…” The authors could introduce right away that DMSO is used as a control, for the different uses, and exemplifies that is wide used as a control or is part of the media component.

- One question rises, why is commonly used the DMSO as part of the media or food components? If various authors have raised concerns about different results obtained before the use of DMSO on their experiment, why is still used? I agree with the authors that is essential to understand the mechanisms and to standardize this, but these are valid questions which could change some results obtained before for others authors.

- On material and methods section: is important to clarify all the component used on this study, mainly because the authors are trying to explain a found a DMSO concentration that does not affect o affect the zebrafish embryos and larvae, so, the chemical and physical characteristics of the water are important to know. The negative control embryos and larvae is the media without DMSO?

- Line: “…In total the experiment was repeated 3 times.” Technical repetition or biological repetition? On different days?

- The E3 solution need to clarify the chemical components or is an industrial solution?

- Result section: “DMSO is a universal solvent for compound that are not miscible in water. An analysis of the number of papers published in the Europe PMC data base using the descriptor DMSO resulted in 475,368 articles published in a period that spanned the years 1963 to 2025. Figure 1.” First, is quite ambiguous to search only the “DMSO” as a descriptor, because, a vast majority of buffers used on different experiments o assays used DMSO as a critical or subcritical chemical component. Second, I do not follow that this information is critical, because the authors are trying to link the use of DMSO on development studies over time.

- Discussion: It could be beneficial for the authors and the lectors that the authors refer to the concentrations that affect the physiological changes. Also, all these DMSO effects at embryological level could be explained at the molecular and biochemical level?

Reviewer #3: 1. Clarify the confirmatory nature and practical relevance of the study.

The authors should explicitly acknowledge in both the Introduction and Discussion that the general toxic effects of DMSO at concentrations above 1% are already well documented in the literature and were therefore expected. The novelty of the present study does not reside in establishing that DMSO is harmful, but rather in the comprehensive and detailed phenotypic characterization of its effects across multiple biological systems, including the heart, notochord, musculature, pigmentation, and motility. Framing the study as a systematic phenotypic mapping will more accurately reflect its scientific value.

In addition, the authors are encouraged to clarify the practical implications of their findings, particularly considering that most researchers are already aware that DMSO concentrations above 1% should be avoided. The manuscript would benefit from a more explicit discussion of potential applications of these data. For example:

Could the phenotypes described here serve as positive controls or reference models for developmental toxicity?

Are there specific scenarios in which concentrations of 1–5% DMSO are still used experimentally despite known risks?

Toxicologists typically avoid exceeding known safety thresholds without justification. Therefore, it would strengthen the manuscript to contextualize when and why such higher concentrations might still be relevant in research.

2. Additional Suggestions:

Include lower concentrations of DMSO (<1%) to help identify subtle or threshold-level effects that may be missed at higher doses.

Incorporate molecular endpoints (e.g., markers of oxidative stress or apoptosis) to better connect morphological changes to underlying mechanisms.

Clearly state the number of embryos (n) analyzed per experimental group in both the Results section and figure legends.

Consider providing a supplementary table with raw or summarized quantitative data (mean ± SD or SEM) for each measured parameter to enhance transparency and reproducibility.

Reviewer #4: The manuscript adds data to an already extensively studied phenomena. In a nutshell the results support the well established notion that DMSO is toxic at concentrations equal to or greater than 1%. It is worth noting that the authors are well aware of this as they cite most of the relevant literature. In my opinion the manuscript shows that the authors have established the conditions to conduct state of the art techniques such as high resolution microscopy and real time video microscopy. Now they can probe more risky hypothesis that produce clearly original results. To mind only comes that they probe concentrations below 1% of DMSO or probe solvents that have similar properties to DMSO. Another option is that they produce a critical review of the current literature, as to my understanding the last one was produced 10 years ago.

6. PLOS authors have the option to publish the peer review history of their article (what does this mean? ). If published, this will include your full peer review and any attached files.

**Do you want your identity to be public for this peer review?** For information about this choice, including consent withdrawal, please see our Privacy Policy .

Reviewer #1: **Yes: ** INTAN SUHANA ZULKAFLI

Reviewer #2: **Yes: ** Dr Alan Briones

Reviewer #3: **Yes: ** Jordana Andrade Santos

Reviewer #4: **Yes: ** Alberto Jose Cabrera Quintero

---

## [Author Response · Author response to Decision Letter 1]

4 Jul 2025

Dear Dr. Hai Xu,

Peer review of our manuscript “DMSO induces major morphological and physiological alterations in zebrafish embryos” (PONE-D-25-27082) was received and according to the Referees, our manuscript has merit but is not suitable for publication as it currently stands. We were requested to submit a revised version of the manuscript addressing the points raised by both reviewers. We have now addressed their criticisms and incorporated their suggestions in a new version of the manuscript. We highlighted the changes throughout the text using yellow color. We also included a point-by-point response to their comments, which were useful for improving data presentation and interpretation. We thank the Referees for their careful and appropriate analysis. The detailed corrections are as follows.

Reviewer #1:

1 - Abstract & introduction: well written & adequate with logical flow.

Methodology:

- Well written, robust methodology, experimental setup is rigorous.

- The details of DMSO used in this study are not provided (i.e., source, grade, quality etc.) despite being the referenced chemical. The preparation of DMSO, how its quality is maintained throughout the course of the experiments, is also not mentioned.

Author’s response: We thank the reviewer for these comments. We now added more information regarding DMSO source, catalog number, purity and preparation. The new text is: “Embryos and larvae were collected, dechorionated and the treatment started at 24 post fertilization (hpf) with increasing concentrations of Dimethyl Sulfoxide (DMSO 1, 2, 3, 4 and 5%, v/v, from Sigma-Aldrich, code D8418, 99.9% purity) up to 72 hpf”.

2 - Results:

- Well written, explanations are thorough. Good quality images used.

- Figures and statistical analyses are comprehensive and effectively illustrate the dose dependent effect of DMSO

- Table 1 is qualitative. It could be further strengthened using quantified values if available (e.g. percentage of changes etc with concomitant p values).

Author’s response: We appreciate this important comment, and we now included values of quantification and p values in the new Table 1.

3 - Discussion:

Well-written, interesting hypothesis on mitochondrial involvement, ALK5 signaling and membrane fluidity. Could be further strengthened by a schematic diagram summarizing the hypothesized mechanism.

Author’s response: We made a schematic diagram summarizing our results and the suggested hypothesized mechanism. The diagram is in the new Figure 11.

4 - Suggest adding future direction on how to validate these hypotheses.

Author’s response: We included future directions on how these hypotheses could be validated in the Discussion section of the manuscript. The new text is: “This hypothesis could be tested with a detailed analysis of possible alterations in mitochondrial function and morphology after DMSO treatment. Distinct fluorescent probes are available to track oxygen consumption and changes in mitochondrial membrane potential, providing information about mitochondrial respiration and membrane permeability. Furthermore, DMSO effects on zebrafish embryos could also be investigated at the molecular and biochemical level through the analysis of the expression of genes and proteins involved in mitochondrial physiology and ALK5 signaling pathway, as well as markers of oxidative stress or apoptosis”.

5 - Suggest commenting a bit more on the long-term effect of DMSO by adding details why somite morphology was unchanged by prolonged DMSO exposure (despite initial effects).

Author’s response: We included comments and hypotheses in the Discussion of the manuscript on why somite morphology was unchanged by prolonged DMSO exposure despite initial effects. The new text is: “Short term treatment with DMSO (24 and 48 hs) induced major changes in somite’s size and shape, but unexpectedly somite morphology was unchanged by prolonged DMSO exposure (6 days). We can hypothesize that muscle cells, among other cells in zebrafish embryos, acquire resistance to DMSO over time. ATP-binding cassette (ABC) transporters, a ubiquitous family of integral membrane proteins, play a significant role in cellular resistance to drugs (Rees et al., 2009). ABC transporters use ATP to translocate substrates across the plasma membrane removing them from the cell. It is possible that alterations in mitochondria by long term treatment with DMSO could alter ATP production and modulate muscle cell resistance to DMSO. Supporting this hypothesis, it has been shown that mitochondrial ATP fuels ABC transporter-mediated drug efflux in cellular chemoresistance (Giddings et al., 2021)”.

Reviewer #2:

The paper entitled “DMSO induces major morphological and physiological alterations in zebrafish embryos” is unique research where the authors analyze the DMSO influence over the development of zebrafish larvae. Overall is an excellent article that shows the importance of being aware of all the components that the media possess. I have some concerns that I point out as follows:

6 - …”optical clarity of the developing embryo…”, the author could use the “transparency” to illustrate what they are trying to say.

7 - Eliminate the word “All” in the line…” All these characteristics…”.

Author’s response: We thank the reviewer for pointing out some parts of the text that could be improved. We now changed these parts in the new version of the manuscript.

8 - The second paragraph, in which the authors present the DMSO problematic, is kind of weird. “…The most important characteristics of a solvent to be used as control, besides the ability to dissolve a given compound, is the lack of effects in the model…” The authors could introduce right away that DMSO is used as a control, for the different uses, and exemplifies that it is widely used as a control or is part of the media component.

Author’s response: We agree with the reviewer that this part of the Introduction was unclear, and we changed according to the reviewer’s suggestion.

9 - One question arises, why is DMSO commonly used as part of the media or food components? If various authors have raised concerns about different results obtained before the use of DMSO in their experiment, why is it still used? I agree with the authors that it is essential to understand the mechanisms and to standardize this, but these are valid questions which could change some results obtained before for other authors.

Author’s response: We included a discussion in the manuscript on why DMSO is still used as part of the media or food components. The new text is: “Finally, the question of why DMSO is still widely used as part of the media if several studies have raised concerns about its use is related to its unique chemical properties. DMSO is a polar aprotic solvent that dissolves both polar and nonpolar compounds and is miscible in a wide range of organic solvents as well as water. Efforts should be made to find new solvents with lower toxicity for developmental biology studies”.

10 - On material and methods section: it is important to clarify all the components used on this study, mainly because the authors are trying to explain a found a DMSO concentration that does not affect or affect the zebrafish embryos and larvae, so, the chemical and physical characteristics of the water are important to know. The negative control of embryos and larvae is the media without DMSO?

Author’s response: We included in the manuscript a detailed description about the DMSO, the E3 solution and water that we used. We also described that the negative control of embryos and larvae that we used was the media without DMSO. The new text is now: “Embryos and larvae were collected, dechorionated and the treatment started at 24 post fertilization (hpf) with increasing concentrations of Dimethyl Sulfoxide (DMSO 1, 2, 3, 4 and 5%, v/v, from Sigma-Aldrich, code D8418, 99.9% purity) up to 72 hpf. DMSO was diluted in E3 solution (5 mM NaCl, 0.17 mM KCl, 0.33 mM CaCl2, and 0.33 mM MgSO4 in distilled water, pH 7.2, which was made in our lab) before each experiment”.

11 - Line: “…In total the experiment was repeated 3 times.” Technical repetition or biological repetition? On different days?

Author’s response: We thank the reviewer for this important comment. We now added this information in the manuscript, as follows: “Six-well plates were used for the experiments. 10 embryos were distributed per well containing 2 mL of solution. Ten technical replicates (embryos with the same treatment within the same plate) and three biological replicates (embryos with the same treatment but performed at a different day and mating) were done (n = 30). Zebrafish embryos and larvae kept in E3 solution without DMSO were considered negative control”.

12 - The E3 solution needs to clarify the chemical components or is it an industrial solution?

Author’s response: As we explained above, we now included the chemical components of the E3 solution (made in our lab) used in the experiments. The new text is: “DMSO was diluted in E3 solution (5 mM NaCl, 0.17 mM KCl, 0.33 mM CaCl2, and 0.33 mM MgSO4 in distilled water, pH 7.2, which was made in our lab) before each experiment”.

13 - Result section: “DMSO is a universal solvent for compounds that are not miscible in water. An analysis of the number of papers published in the Europe PMC database using the descriptor DMSO resulted in 475,368 articles published in a period that spanned the years 1963 to 2025. Figure 1.” First, it is quite ambiguous to search only the “DMSO” as a descriptor, because a vast majority of buffers used on different experiments and assays used DMSO as a critical or subcritical chemical component. Second, I do not follow that this information is critical, because the authors are trying to link the use of DMSO on development studies over time.

Author’s response: We thank the reviewer for questioning this point. We decided to exclude the old Figure 1 with the Europe PMC data from the manuscript. Nevertheless, we included an analysis using both descriptors “DMSO” and “developmental biology”. We believe that now this analysis is more related to our manuscript question. The new text is: “DMSO is an universal solvent for compounds that are not miscible in water. An analysis of the number of papers published in the Europe PMC (https://europepmc.org/) database using the descriptor “DMSO” resulted in 484, 992 articles published (data of search June 01, 2025) in a period that spanned the years 1963 to 2025. Beginning in 2007, there was an exponential increase over time in the number of publications with DMSO. Remarkably, in only one year (2022) more than 45,000 articles with the descriptor DMSO were published, confirming that DMSO has been widely used during the last 60 years. Analysis of the content of these articles showed that DMSO is used in basic biomedical research, cryopreservation, pharmaceutical industry, as well as in developmental biology studies. A search for papers in the Europe PMC using both “DMSO” and “developmental biology” resulted in 15,863 (data of search June 01, 2025) in a period that spanned the years 1971 to 2025”.

14 - Discussion: It could be beneficial for the authors and the lecturers that the authors refer to the concentrations that affect physiological changes. Also, could all these DMSO effects at embryological level be explained at the molecular and biochemical level?

Author’s response: We now added the following new text in the Discussion of the manuscript: “This hypothesis could be tested with a detailed analysis of possible alterations in mitochondrial function and morphology after DMSO treatment. Distinct fluorescent probes are available to track oxygen consumption and changes in mitochondrial membrane potential, providing information about mitochondrial respiration and membrane permeability. Furthermore, DMSO effects on zebrafish embryos could also be investigated at the molecular and biochemical level through the analysis of the expression of genes and proteins involved in mitochondrial physiology and ALK5 signaling pathway, as well as markers of oxidative stress or apoptosis”.

Reviewer #3:

15 - Clarify the confirmatory nature and practical relevance of the study. The authors should explicitly acknowledge in both the Introduction and Discussion that the general toxic effects of DMSO at concentrations above 1% are already well documented in the literature and were therefore expected. The novelty of the present study does not reside in establishing that DMSO is harmful, but rather in the comprehensive and detailed phenotypic characterization of its effects across multiple biological systems, including the heart, notochord, musculature, pigmentation, and motility. Framing the study as a systematic phenotypic mapping will more accurately reflect its scientific value. In addition, the authors are encouraged to clarify the practical implications of their findings, particularly considering that most researchers are already aware that DMSO concentrations above 1% should be avoided. The manuscript would benefit from a more explicit discussion of potential applications of these data. For example: Could the phenotypes described here serve as positive controls or reference models for developmental toxicity?

Author’s response: We thank the reviewer for these important comments. We now added the following new text at the end of the Discussion section of the manuscript: “The novelty of our study resides in a comprehensive and detailed phenotypic characterization of the effects of DMSO across multiple zebrafish biological systems, including the heart, notochord, musculature, pigmentation, and motility. Our study describes systematic phenotypic mapping of the morphological and physiological effects of different concentrations of DMSO in zebrafish embryos and could be expanded to test other solvents that have similar properties to DMSO. The DMSO-associated phenotypes described here could be used as positive controls or reference models for developmental toxicity studies using zebrafish”.

16 - Are there specific scenarios in which concentrations of 1–5% DMSO are still used experimentally despite known risks? Toxicologists typically avoid exceeding known safety thresholds without justification. Therefore, it would strengthen the manuscript to contextualize when and why such higher concentrations might still be relevant in research.

Author’s response: This is an interesting question. We added a comment on this in the Discussion of the manuscript, as follows: “Importantly, in specific scenarios the concentrations of 1-5% DMSO could still be used experimentally in zebrafish embryos despite the described known risks. For example, our data shows that heart beating frequency and pericardium area of 24-hpf zebrafish embryos were affected after 48 hs in the presence of 3-4% DMSO, but 2% DMSO could be used in heart-related studies even though these 3-4% DMSO could affect other organs or systems”.

2. Additional Suggestions:

17 - Include lower concentrations of DMSO (<1%) to help identify subtle or threshold-level effects that may be missed at higher doses.

Author’s response: We now included a new figure (Supplementary Figure 4) in the manuscript with results from new experiments with lower concentrations of DMSO (<1%). We included the following text in the Results section of the manuscript: “Since we found morphological and physiological effects of DMSO treatment in 72 hpf embryos with the concentration of 1%, we decided to test whether concentrations below 1% DMSO could lead to alterations in the locomotion of embryos. Zebrafish embryos were treated with 0.1%, 0.3%, 0.5%, 1% and 2% DMSO and their movements were recorded. The swimming area of embryos was quantified, and no significant alterations were found (Supplementary Material 4), although a dose-dependent tendency of decrease in locomotion was observed”.

18 - Incorporate molecular endpoints (e.g., markers of oxidative stress or apoptosis) to better connect morphological changes to underlying mechanisms.

Author’s response: Thank you for these comments. We included these suggestions in the Discussion of the manuscript: “This hypothesis could be tested with a detailed analysis of possible alterations in mitochondrial function and morphology after DMS

---

## [Decision Letter · Decision Letter 1]

31 Jul 2025

DMSO induces major morphological and physiological alterations in zebrafish embryos

PONE-D-25-27082R1

Dear Dr. Mermelstein,

We’re pleased to inform you that your manuscript has been judged scientifically suitable for publication and will be formally accepted for publication once it meets all outstanding technical requirements.

Kind regards,

Hai O. Xu

Academic Editor

PLOS ONE

Additional Editor Comments (optional):

Reviewers' comments:

Reviewer's Responses to Questions

**Comments to the Author**

1. If the authors have adequately addressed your comments raised in a previous round of review and you feel that this manuscript is now acceptable for publication, you may indicate that here to bypass the “Comments to the Author” section, enter your conflict of interest statement in the “Confidential to Editor” section, and submit your "Accept" recommendation.

Reviewer #1: All comments have been addressed

Reviewer #2: All comments have been addressed

Reviewer #4: All comments have been addressed

2. Is the manuscript technically sound, and do the data support the conclusions?

Reviewer #1: Yes

Reviewer #2: Yes

Reviewer #4: Yes

3. Has the statistical analysis been performed appropriately and rigorously? 

Reviewer #1: Yes

Reviewer #2: Yes

Reviewer #4: Yes

4. Have the authors made all data underlying the findings in their manuscript fully available?

Reviewer #1: Yes

Reviewer #2: Yes

Reviewer #4: Yes

5. Is the manuscript presented in an intelligible fashion and written in standard English?

Reviewer #1: Yes

Reviewer #2: Yes

Reviewer #4: Yes

6. Review Comments to the Author

Reviewer #1: The authors have satisfactorily addressed all comments by the reviewer.

The discussion is strengthened by explaining the potential reasons for DMSO less damaging long-term effects making the manuscript more transparent and robust.

Reviewer #2: (No Response)

Reviewer #4: I have no additional comments. The authors have adressed my previous concerns. Unexpectedly they even included more experiments.

7. PLOS authors have the option to publish the peer review history of their article (what does this mean? ). If published, this will include your full peer review and any attached files.

**Do you want your identity to be public for this peer review?** For information about this choice, including consent withdrawal, please see our Privacy Policy .

Reviewer #1: **Yes: ** intan suhana zulkafli

Reviewer #2: **Yes: ** Dr Alan Briones, PhD.

Reviewer #4: **Yes: ** Alberto Jose Cabrera Quintero

---

## [Editor Report · Acceptance letter]

PONE-D-25-27082R1

PLOS ONE

Dear Dr. Mermelstein,

I'm pleased to inform you that your manuscript has been deemed suitable for publication in PLOS ONE. Congratulations! Your manuscript is now being handed over to our production team.

Kind regards,

on behalf of

Dr. Hai O. Xu

Academic Editor

PLOS ONE